# Investigation of EEG-Based Biometric Identification Using State-of-the-Art Neural Architectures on a Real-Time Raspberry Pi-Based System

**DOI:** 10.3390/s22239547

**Published:** 2022-12-06

**Authors:** Mohamed Benomar, Steven Cao, Manoj Vishwanath, Khuong Vo, Hung Cao

**Affiliations:** 1Department of Electrical Engineering and Computer Science, University of California, Irvine, CA 92697, USA; 2Northwood High School, Irvine, CA 92620, USA; 3Department of Computer Science, University of California, Irvine, CA 92697, USA; 4Department of Biomedical Engineering, University of California, Irvine, CA 92697, USA

**Keywords:** EEG, biometrics, deep learning, Raspberry Pi

## Abstract

Despite the growing interest in the use of electroencephalogram (EEG) signals as a potential biometric for subject identification and the recent advances in the use of deep learning (DL) models to study neurological signals, such as electrocardiogram (ECG), electroencephalogram (EEG), electroretinogram (ERG), and electromyogram (EMG), there has been a lack of exploration in the use of state-of-the-art DL models for EEG-based subject identification tasks owing to the high variability in EEG features across sessions for an individual subject. In this paper, we explore the use of state-of-the-art DL models such as ResNet, Inception, and EEGNet to realize EEG-based biometrics on the BED dataset, which contains EEG recordings from 21 individuals. We obtain promising results with an accuracy of 63.21%, 70.18%, and 86.74% for Resnet, Inception, and EEGNet, respectively, while the previous best effort reported accuracy of 83.51%. We also demonstrate the capabilities of these models to perform EEG biometric tasks in real-time by developing a portable, low-cost, real-time Raspberry Pi-based system that integrates all the necessary steps of subject identification from the acquisition of the EEG signals to the prediction of identity while other existing systems incorporate only parts of the whole system.

## 1. Introduction

Biometrics is a well-valued approach to subject identification for protecting access to a source of information and providing identification without the need for a password or other physical methods [1]. It captures the physiological characteristics of a subject for identification and classification. Biometrics requires measurable physical or behavioral characteristics capable of satisfying the following considerations: universality, distinctiveness, collectability, circumvention, permanence, acceptability, and performance. There are different types of biometrics depending on the physiological characteristic, such as fingerprint, facial structure, iris, and voice, to the less popular ones such as EEG, ECG, deoxyribonucleic acid (DNA), gait, and ear shape [2,3,4]. EEG is an electrophysiological signal that reflects unique cognitive and neurological information of the person; thus, EEG possesses the potential to be used for robust biometrics [1,5,6]. EEG provides a high level of security since it is very difficult to reproduce the EEG patterns of a specific person. In addition, in the event that the person is subjected to a forced EEG reading against his/her will, the EEG biometric system may detect the person’s stress and deny access.

Deep learning-based models have achieved state-of-the-art results in a wide range of clinical applications [7] and biometric identification problems, ranging from phone authentication to bank security systems [8]. The use of DL models for biometric identification has been increasing in recent years. Within the field of EEG biometrics, DL models have been leveraged to improve the accuracy of EEG biometric identification systems.

Most Machine Learning (ML) projects are performed on ordinary computers or Graphics Processing Units (GPUs) because of the computing power they offer. However, since the purpose of this project is to develop a portable Internet of Things (IoT) system, it is not feasible to use common computers. In this work, we use Raspberry Pi as the core, which is compact, portable, and Wi-Fi capable. Raspberry Pi is widely used for portable IoT applications because of its versatility. The Raspberry Pi has been used in a number of projects for a variety of purposes, including monitoring the health status of patients [9], a security system [10], and a testing system device [11] to name a few. The Raspberry Pi has also been used for projects based on EEG signals. For example, in [12], EEG is used to control a car, and, in [13], it is used to monitor the depth of anesthesia. The purpose of this work is to develop a first approximation to a fully functional portable system for subject identification that uses trained DL models to process the signals.

This paper is organized as follows: The data description is presented in Section 2. In Section 3, we describe the software implementation for the subject identification using EEG signals. Section 4 shows the hardware implementation of the system. Section 5 reports the results. Finally, the discussion and conclusion will be drawn in Section 6.

## 2. Dataset Description

The BED (Biometric EEG Dataset) [14] is a dataset specifically designed to test EEG-based biometric approaches that use relatively inexpensive consumer-grade devices. The dataset includes EEG responses from 21 subjects to 12 different stimuli, which were broadly divided into four different types, namely affective stimuli, cognitive stimuli, visual evoked potentials, and resting-state. Each stimulus contains data across three different chronologically disjointed sessions. Fourteen-channel EEG signals containing the channels—AF3, F7, F3, FC5, T7, P7, O1, O2, P8, T8, FC6, F4, F8, and AF4 as shown in Figure 1—were collected at a sampling rate of 256 Hz. The BED dataset includes the raw EEG recordings with no preprocessing and the log files of the experimental procedure, in text format. The EEG recordings were segmented, structured, and annotated according to the presented stimuli, in Matlab format. The BED dataset also includes Autoregression Reflection Coefficients (ARRC), Mel-Frequency Cepstral Coefficients (MFCC), and Spectral (SPEC) features that were extracted from each EEG segment. In this work, however, only raw EEG recordings without any manually extracted features were used. Out of the 12 different stimuli, we only used EEG signals recorded during the Rest-Closed (RC) Stimulus since the action of “closing your eyes” is an easy and natural action and could be best replicated in a real-world scenario for biometric applications with ease. Having stimuli in the experimental procedure will require additional devices to present them to the individuals, which in turn will increase the complexity of the whole setup and practical use in the real world.

The BED dataset was simulated using our real-time Raspberry Pi-based system as a proof of concept to test the feasibility of this work for any real independent practical use. The simulated data acquisition in our Raspberry Pi-based system is performed by an analog-to-digital converter (ADC) that reads the analog output from the digital-to-analog converter (DAC) that receives the input from the original saved BED dataset. More details will be discussed in Section 4.

## 3. Software Implementations

In this section, we describe the data preprocessing steps and the deep learning models used. We also discuss how these models will be evaluated.

### 3.1. Data Processing

#### 3.1.1. Preprocessing of EEG Signals

The preprocessing step of EEG signals is a crucial step in the DL pipeline because of its impact on the EEG analysis process. Without the preprocessing step, there may be noisy data and artifacts that can mask distinct features in the EEG signals. This can cause the model to have a harder time distinguishing between relevant EEG features, resulting in poorer performance of the model. In addition, we must pay attention to the quality of the preprocessing step as it can introduce unwanted artifacts if the early stages of the pipeline are not properly addressed. For example, although ordinary average referencing improves the signal-to-noise ratio, noisy channels which depend on the reference can contaminate the results.

Figure 2 shows the different steps of EEG signal preprocessing. The well-known preprocessing technique—PREP pipeline [15]—introduces specific important functionality for referencing the data, removing line noise, and detecting bad channels in order to deal with noisy channel-reference interactions. The PREP pipeline also removes artifacts such as muscle movement, jaw clenching, and eye blinking. The pipeline consists of various steps. First, the signal is filtered using a 1 Hz high pass filter followed by line noise removal using a notch filter at 60 Hz. In addition, finally, the signal is robustly referenced with respect to an estimate of the true mean reference, thereby enabling the detection of faulty channels. These channels are then interpolated relative to the same reference. We then use a lowpass filter of 50 Hz and divide the EEG data into overlapping epochs with an overlap rate of 90 percent. Finally, we standardize the EEG signals for each channel using StandardScaler. Figure 3 depicts an EEG epoch before and after preprocessing.

#### 3.1.2. Deep Learning Implementation

The deep learning models used for subject identification are based on ResNet [16], an extension of the neural network into internal structures that add direct connections to the internal residual blocks to allow the gradient to flow directly through the lower layers; Inception [17], a convolutional neural network architecture that executes multiple operations with multiple filter sizes in parallel to avoid facing any compensation and allows the network to automatically extract relevant features from the time series; and EEGNet [18], a compact convolutional neural network that has been designed to build an EEG-specific model as it includes concepts and tools specific to EEG signals such as feature extraction and optimal spatial filtering to reduce the number of trainable parameters. The utilization of these three DL models was based on the fact that they are state-of-the-art models that have achieved good results for various other applications. For example, ResNet and InceptionTime were designed for Time Series Classification. EEGNet is a DL model that was designed for EEG-based Brain-Computer Interfaces. As a result, we wanted to examine the use of these models for the EEG Biometrics application.

Modifications for ResNet include adding additional residual blocks, to understand whether a more complex model that could extract complex features would perform better. For the Inception model, additional dropout layers and inception blocks were added. The activation function was changed from Rectified Linear Unit (ReLU) to Exponential Linear Unit (ELU) as the model was overfitting. For the EEGNet model, we fine-tuned the length of temporal convolution in the first layer and the number of channels after trial and error. Other modifications, such as adding GlobalAveragePooling2D layer, varying the dropout rate from 40 percent to 60 percent, and rearranging the order of layers, did not significantly improve the model performance.

In addition, within all our models, we added a callback function that reduces the learning rate based on the training loss. Specifically, we added the hyperparameter called patience, which is the number of epochs of non-decreasing loss values that the model runs before reducing the learning rate by half.

Because we want to check the feasibility of EEG-based Biometric identification over long periods of time, these presented models were trained using the first two weeks of the three chronologically disjoint sessions in the BED dataset. The third week of data was used for testing the trained model.

Once the models were trained, we had their hyperparameters, including learning rate, batch size, number of filters, kernel size, and number of epochs fine-tuned. The hyperparameter learning rate was set to 0.003 for the ResNet model and 0.009 for the Inception and EEGNet models. In addition, the number of epochs has been modified, 150 for Inception and 400 for ResNet. Figure 4, Figure 5 and Figure 6 briefly show the overall architectures of our modified Resnet-based, Inception-based, and EEGNet-based framework used in this work.

### 3.2. Model Evaluation

The model was evaluated using the third session of the three chronologically disjoint sessions provided in the dataset. The trained model would take one EEG segment from the testing dataset at a time, and output the prediction of which person it belongs to. Once all the EEG segments in the testing dataset have been predicted, we would use the confusion matrix to evaluate the model performance. The evaluation of the machine learning algorithms was carried out by comparing the most relevant indices for the prediction of the subjects [19]. The accuracy of the models calculates the ratio of correct predictions over the total number of instances evaluated. Precision is used to measure the positive patterns that are correctly predicted from the total predicted patterns in a positive class. Recall is used to measure the fraction of positive patterns that are correctly classified. *F*1 *Score* is calculated to represent the harmonic mean between recall and precision values. Finally, the *Precison* vs. *Recall* (P–R) curves are obtained to provide a graphical representation of the DL models’ performance.
(1)$$Precision=\frac{TP}{TP\;+\;FP}$$
(2)$$Recall=\frac{TP}{TP\;+\;FN}$$
(3)$$F1\;Score=2\cdot \frac{Precision\cdot Recall}{Precision\;+\;Recall}$$
$$\begin{array}{cccc} TP: & \mathrm{True}\;\mathrm{Positive} & FP: & \mathrm{False}\;\mathrm{Positive}\\ TN: & \mathrm{True}\;\mathrm{Negative} & FN: & \mathrm{False}\;\mathrm{Negative} \end{array}$$

## 4. Hardware Implementation

In this section, we describe the different parts of the system which are responsible for the generation and acquisition of EEG data and its processing for subject identification. In addition, finally, we present the system validation and its performance.

### 4.1. Data Acquisition

In this experimental work, data acquisition is simulated by using data from an existing database as mentioned in Section 2. Real-time EEG acquisition is in the scope of the future where we intend to implement the best deep learning model obtained from this study along with real-time signal acquisition for biometric application. Keeping in mind our goals for future work and existing time constraints, we decided to simulate analog EEG input in this work as mentioned later in this section.

The system consists of a Digital-to-Analog Converter (DAC) in charge of converting the stored data to analog signals mimicking real EEG acquisition scenarios [20]. The DAC is the 12-bit MCP4725 chip with an Inter-Integrated Circuit (I2C) communication bus [21]. Since the converter is a 12-bit converter, the range of digital values that can be converted is from 0 to 4095. Therefore, before converting the EEG signals stored in the memory of the Raspberry Pi to analog signals, the data were transformed by scaling each value to the range from 0 to 4095. The design includes an electronic loop with an Analog-to-Digital Converter (ADC) that transforms the analog EEG signal from the subjects—in this case, the DAC, to a digital signal for its processing by machine learning algorithms. The ADC is the 10-bit MCP3008 chip with Serial Peripheral Interface (SPI) [22]. The analog data are converted to digital signals by 10-bit architecture, so the digital data range is from 0 to 1023. As a result, we simulated the acquisition of EEG signals by means of an electronic loop between an ADC and a DAC using the data from the BED dataset. Figure 7 illustrates the complete hardware of the system and its connections.

### 4.2. Data Processing

The acquired signals are processed by the system controller, which is based on the Raspberry Pi 4 Model B 4 GB RAM, which performs the tasks of capturing the EEG signals, pre-processing them using a pipeline, and processing them through machine learning algorithms for the identification of the subject. The Raspberry Pi is a single-board computer based on Linux [23]. The programming language used is Python [24] as it is the most suitable for machine learning applications owing to a large number of available libraries. Task management is carried out by threads that run in parallel. The first thread consists of constant EEG data acquisition by the ADC, which is stored in a buffer for their processing, and the second thread is in charge of the pre-processing technique and the classification in real time of the samples stored in the buffer.

The communication between the user and the Raspberry is established using the SSH protocol [25]. Therefore, the result of the subject identification can be obtained in other edge devices, such as a PC or a server. Here, the result is displayed on the terminal indicating the identification of the subject.

### 4.3. System Validation

For test management, the Raspberry Pi handles another thread responsible for converting the dataset into EEG signals that are acquired by the ADC. To avoid data overlapping in the process of writing—DAC—and reading—ADC—the samples, a synchronization method is used between the two threads. Synchronization involves multiple threads efficiently waiting for each other to finish tasks. Specifically, the ADC thread waits to read the data until the DAC finishes writing the data, and the DAC waits to write new data until the ADC finishes reading the previous one. The ADC thread stores the read samples in a buffer, from which the classification thread reads them. Figure 8 illustrates the block diagram of the algorithm of the system software.

## 5. Results

### 5.1. Dataset Generator Efficiency

Generating analog EEG signals from a previously saved dataset by means of a DAC may generate an error between the original and the generated data. This error may be due to the mismatch of the bit architecture between the two converters or due to the error generated by each converter when generating the analog or digital signal in each case. Therefore, to analyze the efficiency of the dataset regeneration, the Mean Squared Error (*MSE*) between the two datasets was calculated using (4):(4)$$MSE=\frac{1}{N}\cdot \sum_{n=1}^{N}{\left(y_{i}-{\hat{y}}_{i}\right)}^{2}$$
where *N* denotes the total number of samples in the dataset, $y_{i}$ denotes the original dataset sample, and ${\hat{y}}_{i}$ denotes the sample generated by the DAC. The MSE between the original dataset and the dataset generated by the DAC is 27.57, which represents 0.67% of the dataset range.

### 5.2. Model Efficiency

Table 1 shows the performance of the three DL models used. We evaluated the models using four metrics: accuracy, *F*1 *score*, precision, and recall. Using the preprocessing PREP pipeline, the EEGNet-based model obtains the best accuracy of 86.47% followed by the Inception-based model with an accuracy of 70.18% and the ResNet-based model with an accuracy of 63.21%. In addition, we obtained for each DL model the processing time of one sample.

Figure 9 illustrates the precision vs. recall curves for different DL models to provide a graphical representation of each model’s efficiency. It can be seen that the P–R curve of the EEGNet model indicates the best results with higher precision and recall values.

## 6. Discussion and Conclusions

In this work, we investigated EEG-based subject identification using a consumer-grade EEG acquisition device (Emotiv EPOC+ headset) using several deep learning models that are based on ResNet, Inception, and EEGnet. We used our deep learning models on the BED dataset, an existing online EEG dataset and came out with a good deep learning EEGNet model at an accuracy of 86.74%. The high accuracy shows a promising future for EEG-based subject identification over long periods of time.

The high performance of the EEGNet-based model compared to the Inception-based model and ResNet-based model may be due to its use of Depthwise and Separable Convolutions allowing the construction of an EEG-specific model incorporating well-known EEG features.

We plan to validate these deep learning models on more training datasets once we collect them ourselves.

To better examine the performance of our deep learning models, we compared our results with previous works that use the same BED dataset [14] on subject identification tasks. The comparison is shown in Table 2. Compared to their best accuracy of 47.70%, 83.51%, and 29.69% respectively, our work on the EEGNet-based model demonstrates a higher accuracy of 86.74%. Here, ACC refers to accuracy and CCR refers to Correct Classification Rate.

As for the hardware design of this work, a proof of concept EEG-based real-time identification system using Raspberry Pi is demonstrated that integrates all the components needed for the practical use of a real subject identification system. The execution of the thread parallelism has been performed without any problem, and all the threads are synchronized to achieve their tasks without any errors. The MSE is low with respect to the range of the data conveying high similarity between the original dataset and the data generated by the DAC, which demonstrates the reliability of the results obtained during the model evaluation stage.

Finally, Raspberry Pi provides portability and allows the system to be integrated into a network; hence, the information is accessible from anywhere. We plan to develop a system for EEG acquisition and integrate it with the current Raspberry Pi-based system. The development of an acquisition system will eliminate the use of converter DAC and the need for a pre-stored dataset. The motivation for developing our device comes from the comparison of the different commercially available products. The devices found were developed only for acquisition purposes or for more complex applications and most of them are very costly. For this reason, we are motivated to develop our own end-to-end, low-cost, real-time device involving EEG acquisition from minimum necessary channels for EEG biometrics, preprocessing, feature extraction, and subject identification. In the future, to improve our device, we will perform a deeper analysis of the hardware by studying the power and time consumption and alternative technology for minimization of cost and size of the device.

## Figures and Tables

**Figure 1 sensors-22-09547-f001:**
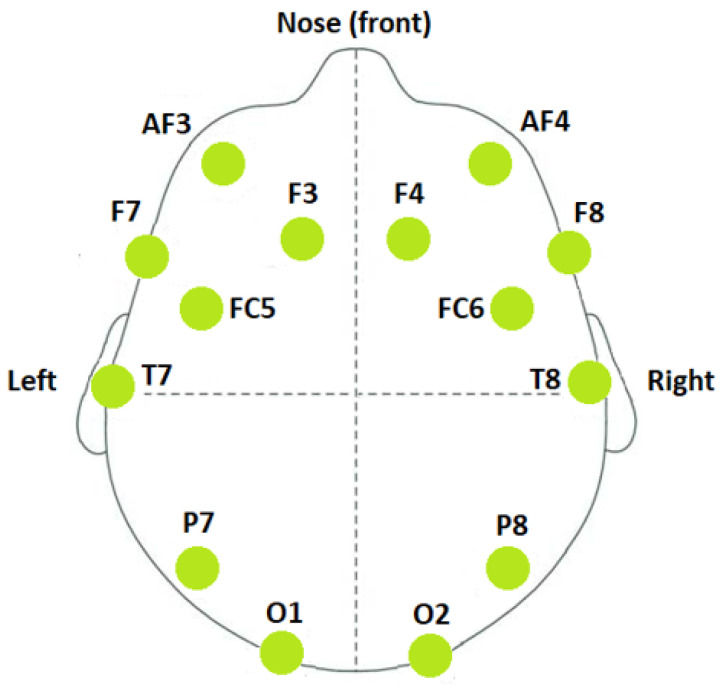
Electrode montage.

**Figure 2 sensors-22-09547-f002:**
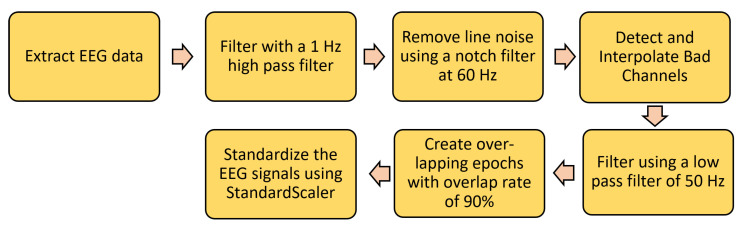
EEG preprocessing steps.

**Figure 3 sensors-22-09547-f003:**
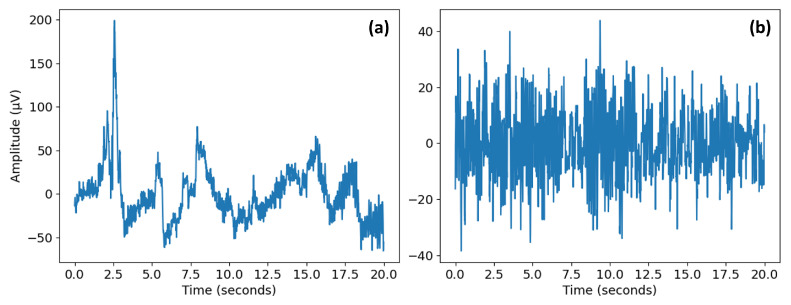
EEG epoch (**a**) raw data; and (**b**) preprocessed epoch.

**Figure 4 sensors-22-09547-f004:**
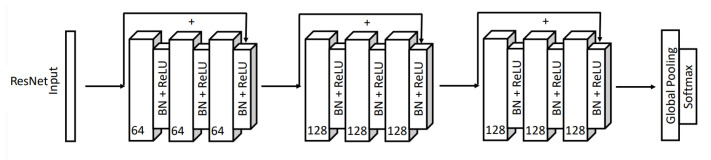
ResNet model architecture.

**Figure 5 sensors-22-09547-f005:**
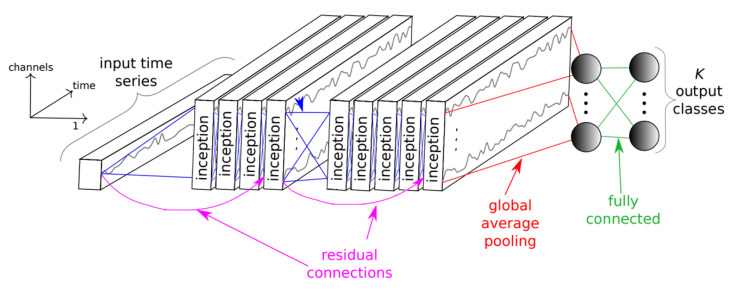
Inception model architecture.

**Figure 6 sensors-22-09547-f006:**
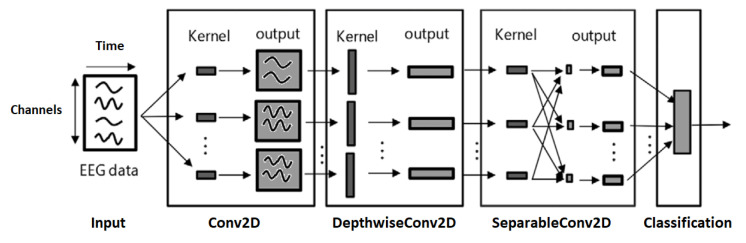
EEGNet model architecture.

**Figure 7 sensors-22-09547-f007:**
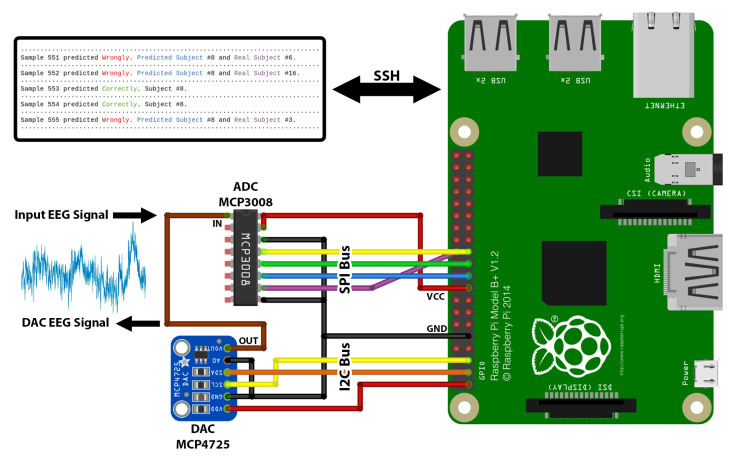
System hardware setup.

**Figure 8 sensors-22-09547-f008:**
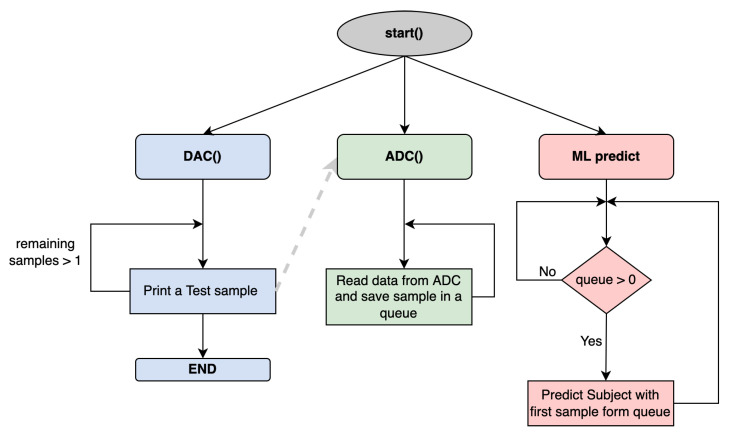
Real-time acquisition algorithm.

**Figure 9 sensors-22-09547-f009:**
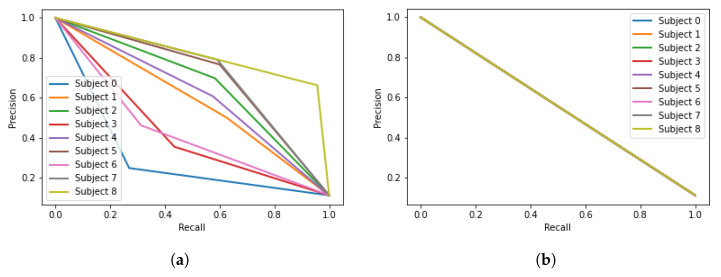
P–R Curves for DL models (**a**) ResNet model; (**b**) Inception model; (**c**) EEGNet model.

**Table 1 sensors-22-09547-t001:** Comparison of performance between Deep Learning Models.

Model	ResNet-Based	Inception-Based	EEGNet-Based
*Accuracy* (%)	63.21	70.18	86.74
*F*1 *Score* (%)	61.55	69.87	86.69
*Precision* (%)	63.90	74.22	89.13
*Recall* (%)	63.21	70.14	86.68
*Processing Time* (ms)	9.35	10.5	10.28

**Table 2 sensors-22-09547-t002:** Comparison with existing works.

Paper	Data Used	Metrics
BED: A New Data Set for EEG-Based Biometrics [14]	MFCC, ARRC, and SPEC features for all 12 stimulus	ACC: 47.79%
Multi Channel EEG Based Biometric System with a Custom Designed Convolutional Neural Network [26]	Raw EEG using 4 channels of RC Stimulus	CCR: 83.51%
Single-channel EEG-based subject identification using visual stimuli [27]	MFCC, ARRC, and SPEC features of all stimulus for each channel	ACC: 29.69%
Our proposed best model based on EEGNet	Raw EEG of RC Stimulus	ACC: 86.74%

## Data Availability

https://doi.org/10.5281/zenodo.4309471.

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
