# Peer review of "Investigation of EEG-Based Biometric Identification Using State-of-the-Art Neural Architectures on a Real-Time Raspberry Pi-Based System"

_sensors, 2022, doi:10.3390/s22239547_

Round 1
Reviewer 1 Report
General comments:
The manuscript proposes a deep learning model based on human brainwave signals (EEG), and after comparing the PR values of ResNet-based, Inception-based and EEGNet-based, three DL models, the advantages of high recognition accuracy and recall of the EEGNET model are derived, which is of reference significance for deep learning application research.
Special comments:
(1) Line 144, does the # symbol of the formula appear garbled?
(2) Line 219, it would be more convincing to give the PR curves of the three models for comparison
(3) The manuscript uses a large amount of space in signal data processing, DL models, and the arrangement of Raspberry Pi, but less space in the results and conclusions, only comparing the recognition accuracy of different models and the recall rate, could the results be described in more detail?
(4) Can you give the detection speed of the hardware Raspberry Pi?
Reviewer 2 Report
The paper explores the use of state-of-the art deep learning models for EEG-based subject identification tasks.
Models as ResNet, Inception, and EEGNet are aligned to classify EEG recordings from 21 individuals. In addition it is demonstrated how to perform EEG biometric tasks in real-time by developing a portable, low-cost, real-time Raspberry Pi-based system.
All in all, the paper is well written. It provides enough background information to understand the conducted experiments. The main goal is clearly outlined. The authors also put special effort in explaining the data preparation processes.
Nevertheless, there are some issues I want to address:
Are the hyperparameters of the presented models the same as the original model? What about the training process? It is only talked about the prediction process and not about the training process of the models. What about the very small amount of data? Please write a small section that explains how the models were trained.
You are demonstrating how to perform the biometric tasks on a Raspberry Pi-system but you don't provide the source code to reproduce the experiments. Is it possible to do that?
Round 2
Reviewer 1 Report
General comments:
The manuscript proposes a deep learning model based on human brainwave signals (EEG), and after comparing the P-R values of ResNet-based Inception-based EEGNet-based, three DL models, the advantages of high recognition accuracy and recall of the EEGNET model are derived. After the first revision, we can see that the authors have further refined the presentation and structure of the paper.
Special comments:
(1) It can be seen that the EEGNet-based DL model has higher ACC and better recognition results when compared with the other two DL models, but the difference in processing speed between these three models is not reflected in the paper, which is also an important indicator to evaluate the advantages and disadvantages of DL models.
(2) It would be more convincing to give the P-R curves of these DL models to compare the arguments.
Reviewer 2 Report
Dear Authors,
I am very grateful for the detailed answers to my questions. However, I don't understand why you are explaining to me that you are using completely different hyperparameter for the models as one would suggest, but not adding this facts to your paper. How should it be possible for other researchers to reproduce and verify your results?
Please add this information to your paper.
